# P2P-O: A Purchase-To-Pay Ontology for Enabling Semantic Invoices

Michael Schulze[1,2,3], Markus Schröder[1,2], Christian Jilek[1,2], Torsten Albers[3], Heiko Maus[2], and Andreas Dengel[1,2]

[1] Computer Science Department, Technische Universität Kaiserslautern, Germany
[2] Smart Data & Knowledge Services Department, Deutsches Forschungszentrum für Künstliche Intelligenz GmbH (DFKI), Kaiserslautern, Germany
[3] Development Department, b4value.net GmbH, Kaiserslautern, Germany
{firstname.lastname}@dfki.de

**Abstract.** Small and medium-sized enterprises increasingly adopt electronic invoices and digitized purchase-to-pay processes. A purchase-to-pay process begins with making a purchase order and ends with completing the payment process. Even when organizations adopt electronic invoices, knowledge work in such processes is characterized by assimilating information distributed over heterogeneous sources among different stages in the process. By integrating such information and enabling a shared understanding of stakeholders in such processes, ontologies and knowledge graphs can serve as an appropriate infrastructure for enabling knowledge services. However, no suitable ontology is available for current electronic invoices and digitized purchase-to-pay processes. Therefore, this paper presents P2P-O, a dedicated purchase-to-pay ontology developed in cooperation with industry domain experts. P2P-O enables organizations to create semantic invoices, which are invoices following linked data principles. The European Standard EN 16931-1:2017 for electronic invoices was the main non-ontological resource for developing P2P-O. The evaluation approach is threefold: (1) to follow ontology engineering best practices, we applied OOPS! (OntOlogy Pitfall Scanner!) and OntoDebug; (2) to evaluate competency questions, we constructed a purchase-to-pay knowledge graph with RML technologies and executed corresponding SPARQL queries; (3) to illustrate a P2P-O-based knowledge service and use case, we implemented an invoicing dashboard within a corporate memory system and thus enabled an entity-centric view on invoice data. Organizations can immediately start experimenting with P2P-O by generating semantic invoices with provided RML mappings.

**Keywords:** Semantic invoice · E-procurement · Purchase-to-pay process · Enterprise knowledge graph · Corporate memory · RML

## 1 Introduction

Small and medium-sized enterprises (SMEs) increasingly adopt electronic invoices and move towards digitizing their purchase-to-pay processes. A purchase-to-pay or procure-to-pay process begins with making a purchase order and ends

with completing the payment process [19]. In such processes, invoice processing is an ubiquitous task [19]. This requires besides information on invoices, also information on other documents, such as delivery notes, credit notes or reports on service provisions. Also frequently needed is background information from various data collections, such as suppliers, product catalogs or purchase-to-pay policies. Therefore, this kind of knowledge work is characterized by searching and assimilating information distributed over heterogeneous sources among different stages in the process. Jain & Woodcock [19] estimate that 21% of tasks in the field of invoice processing will be hard to process automatically. Consequently, even when SMEs adopt electronic invoices, human effort in purchase-to-pay processes will continue to be essential.

To assist knowledge workers in such digitized purchase-to-pay processes, knowledge graphs [9] can serve as an appropriate infrastructure for enabling knowledge services by integrating distributed and heterogeneous information from document-based and other data sources. We envision a personal "information butler" [6], who is able to proactively deliver pertinent information depending on a given work context [21]. In purchase-to-pay processes, this context might be a task involving verification of a corrective invoice based on an initial invoice, a purchase order and reports on service provisions. Having a purchase-to-pay knowledge graph also enables the integration into a knowledge description layer of a corporate memory [1]. This provides an appropriate infrastructure for knowledge services embedded into the office environment of daily work [24].

In line with the definition of a knowledge graph suggested by Ehrlinger & Wöß [9], we see an ontology as an inherent part of a knowledge graph. However, to the best of our knowledge, there is no purchase-to-pay ontology available suiting our requirements and goals. These are as follows: describing and interrelating information on current electronic invoices, relating invoices to a corresponding purchase-to-pay process and adhering to industry standards. Related and established ontologies in the field, such as the Financial Industry Business Ontology (FIBO) [3] or the GoodRelations Ontology [17], have a different focus and thus do not provide sufficient vocabulary to meet these requirements. Therefore, this paper presents P2P-O, a dedicated purchase-to-pay ontology.

P2P-O is developed in cooperation with industry domain experts and aligned with the core invoice model of the European Standard EN 16931-1:2017 (EN16931) [10]. Thus, electronic invoices can be upgraded to semantic invoices, which we define as invoices following linked data principles. Besides enabling knowledge services, semantic invoices also enable new kinds of queries. This is evident in the case of incorporating linked open data[4] in federated queries, thus allowing SMEs, for example, to filter which of their products are sold in cities with more than 50.000 inhabitants. Adoption of e-invoices is also associated with positive social and financial consequences. It is estimated that the adoption reduces one million tones of $CO_2$ emissions a year[5]. Also, it helps with reducing the VAT gap

---

[4] `www.lod-cloud.net`

[5] `https://eur-lex.europa.eu/LexUriServ/LexUriServ.do?uri=COM:2010:0712:FIN:en:PDF`

resulted from tax fraud and tax evasion [22]. Financial resources that could be freed up for society. Additionally, it is estimated that in the European Union the adoption can save up to 0.8 % of the gross domestic product (GDP) [4,10]. This is in particular due to resulted process efficiency [10]. By providing added value and incentives in form of semantic knowledge services, we also aim to increase adoption rates, especially those from SMEs because their rate is only half the rate of big enterprises (22 %)[6].

The rest of the paper is structured as follows. The next section introduces purchase-to-pay processes and the European Standard EN16931 as the main non-ontological reused resource. Section 3 presents the ontology and describes the developing and modeling process. Section 4 elaborates on the evaluation approach and on an example use case, and Section 5 covers related resources. We conclude with Section 6 and provide an outlook on future work.

## 2   Foundations

### 2.1   Purchase-To-Pay Processes & Electronic Invoices

Figure 1 depicts a simple instance of a purchase-to-pay process that starts with the sending of a purchase order from the buyer to the seller. After delivering the requested goods or providing the services, the seller sends an electronic invoice to the buyer. In the case of the process in Figure 1, a dispute is depicted and thus, finally, a credit note sent to the buyer. Because P2P-O's focus is on electronic documents in such purchase-to-pay processes, Figure 1 leaves out the actual payments made by the participants as well as the physical exchange of goods.

In practice, purchase-to-pay processes can take on more diverse and complex forms. For instance, instead of sporadic purchase orders (Fig. 1), processes can be periodic based on a contract. Also, despatch and receiving advice documents or service provision documents can be part of purchase-to-pay processes. For a more detailed overview, we kindly refer the reader to the European Standard EN-16931 [10]. In addition, with respect to the participants in purchase-to-pay processes, buyer, seller, receiver, payee and the respective taxable persons do not necessarily have to be the same [10]. Summarizing, purchase-to-pay processes are characterized by numerous heterogeneous documents, especially in SMEs where ERP Systems are often missing, as well as by diverging processes.

### 2.2   The Core Invoice Model in EN 16931-1:2017

In the European Standard EN 16931-1:2017 [10], the CEN-CENELEC Management Centre introduces the *core invoice model*. We reused this standard because of its general approach: the model specifies 161 core information elements for electronic invoices that are sufficient for most transactions. Examples for information elements are the invoice number or the buyer name. In transactions where the core invoice model is not sufficient, it can be extended.

---

[6] https://eur-lex.europa.eu/LexUriServ/LexUriServ.do?uri=COM:2010:0712:FIN:en:PDF

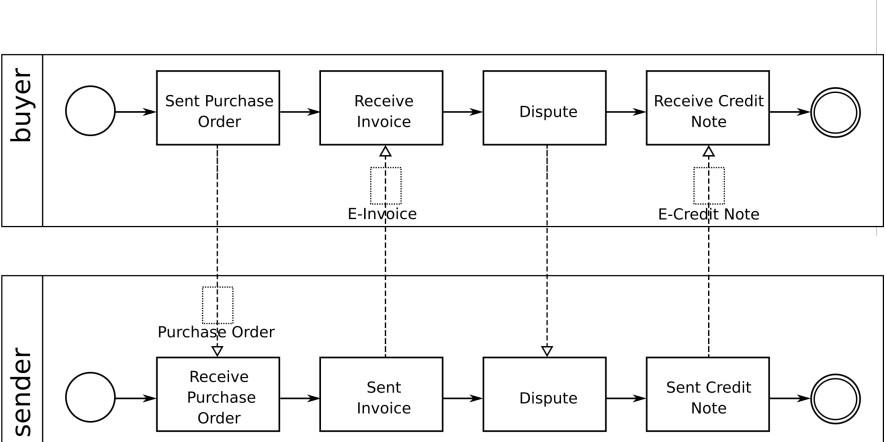

Fig. 1: Example of a simplified purchase-to-pay process with a dispute

In the model, information elements are organized hierarchically at different levels. On the first level (document level), some elements are not further divided (e.g. the invoice number). Higher levels can group further information elements together. For example, information elements with respect to the seller are grouped at the second level (e.g. seller name) and they can be further divided to a third level (e.g. seller postal address). Information elements in EN16931 are depicted in a tabular form with the following columns:

**ID:** identifier, e.g. BT-1 for the information element "invoice number"
**Level:** level of the information element, e.g. + for the first level
**Cardinality:** cardinality of the information element, e.g. 1..n
**Business Term:** name, e.g. "invoice number"
**Description:** further details about the information element
**Usage Note:** notes about the practical usage of the information element
**Requirement ID:** the particular requirement specified in EN16931 which is addressed by the information element, e.g. R56
**Semantic Data Type:** data type, e.g. Text, Identifier or Code

This list will be referred when the modeling process in Section 3.2 is covered. E-invoices cannot be considered in isolation from the respective purchase-to-pay process as its context. Therefore, EN16931 specifies 12 kinds of purchase-to-pay processes that are supported by the invoice model without any extensions. [10]

## 3  P2P-O: The Purchase-To-Pay Ontology

### 3.1  Methodology of the Developing Process

Because EN16931 [10] was the main non-ontological resource for developing P2P-O, we followed the NeOn methodology [31] since it provides established guidelines for this exact scenario. Additionally, we incorporated advice from Grüninger

Table 1: Excerpt of competency questions for P2P-O.

| Identifier | Competency Question |
| --- | --- |
| C1 | What is the reference number of an invoice? |
| C2 | What is the total amount without value added tax of an invoice? |
| C3 | Who is the seller on an invoice? |
| C4 | Who is the buyer on an invoice? |
| C5 | What are items listed on the invoice? |
| C6 | What is the address of the buyer? |
| C7 | To what price and quantity was an item on an invoice purchased? |
| C8 | What are attributes of an item on an invoice? |
| C9 | Which organizations purchased an item? |
| C10 | Which organizations sold an item? |
| C11 | To what sort of purchase-to-pay processes does a document belong to? |
| C12 | What are the documents in a purchase-to-pay process? |
| C13 | Which items on invoices have the colors red and blue? |
| C14 | To which addresses an organization ordered an item? |
| C15 | Which and how many items an organization sold in cities with more than 50 000 inhabitants? |

& Fox [15], Hitzler et al. [18] and McDaniel & Storey [25]. For publishing P2P-O, we followed FAIR principles [32]. The iterative process for developing P2P-O is depicted in Figure 2. At first, we specified requirements, scope and competency questions together with domain experts from the TRAFFIQX network[7] (examples in Tab. 1). The set of competency questions has been derived from the requirement specification in EN16931 and has been then enriched. Classes and properties for potential use have subsequently been derived from EN16931 [10] until a conceptual model was achieved. Lessons learned from conversations with experts were, for example, which elements on invoices are frequently used and how P2P-O needs to be designed to allow for common extensions.

To reuse ontologies and to incorporate them into P2P-O, we applied, in addition to guidelines in the NeOn methodology [31], the validation process for ontologies suggested by McDaniel & Storey [25]. Accordingly, we verified the adequacy of ontologies based on our requirements, and we assessed them by means of OOPS! (OntOlogy Pitfall Scanner!) [28] and OntoDebug [30]. These tools have also been employed for P2P-O's evaluation.

### 3.2   General Modeling Process

This section describes the general modeling process with the *core invoice model* [10] as its basis. More detailed modeling aspects are addressed in the respective ontology modules in Section 3.3. Each column of the core invoice model (Sec. 2.2) has been implemented as follows. For the ID of information elements, we introduced the annotation property *seeEN16931-1-2017* so that resources in

---

[7] https://www.traffiqx.net/en/about-us

P2P-O are linked to information elements in EN16931. This way, it is also possible to query these elements to see how they are modeled in P2P-O. As a result, modeling decisions can be traced from the original invoice model to the ontology and backwards. For the "Business Term" in EN16931, *rdfs:label* is used and for the "Description"-column *rdfs:comment*. Usage Notes, however, are modeled with the annotation property *usage note* from FIBO [3]. Cardinality statements were encoded with OWL class restrictions. For the data type *Text* in EN16931, `xsd:string` was used. This was also used for the type *Code* because it only consists of one text field [10]. However, for the identifier datatype in EN16931 also information regarding the identifier scheme and its version is needed. Therefore, we modeled it as a dedicated class *Identifier* rather than as a property like in Schema.org [16] or DCMI Metadata Terms [5].

### 3.3   Ontology Description

P2P-O comprises seven modules, 54 classes, 169 properties and 1438 axioms. Table 2 summarizes reused ontologies and vocabularies. An important requirement for reusing ontologies was that a permissive license has been specified. To not clutter P2P-O, only selected statements are reused instead of importing entire ontologies. This was especially problematic in the case of FIBO [3] due to long import chains. Figure 3 illustrates an excerpt of P2P-O's schema which will be referred at appropriate places in the following remarks on the various modules.

Table 2: Reused ontologies and vocabularies in P2P-O.

| Prefix | Namespace | Source |
|---|---|---|
| vcard | http://www.w3.org/2006/vcard/ns#Address | [40] |
| fibo-fnd-.. | https://spec.edmcouncil.org/fibo/ontology/FND/.. | [3] |
| omg | https://www.omg.org/spec/LCC/Countries/CountryRepresentation/ | [27] |
| foaf | http://xmlns.com/foaf/0.1/ | [33] |
| org | http://www.w3.org/ns/org# | [39] |
| xsd | http://www.w3.org/2001/XMLSchema | [36] |
| ontodebug | http://ainf.aau.at/ontodebug# | [30] |
| skos | http://www.w3.org/2004/02/skos/core# | [34] |
| dcterms | http://purl.org/dc/terms | [5] |
| owl | http://www.w3.org/2002/07/owl# | [35] |
| rdf | http://www.w3.org/1999/02/22-rdf-syntax-ns# | [37] |
| rdfs | http://www.w3.org/2000/01/rdf-schema# | [38] |

**Module *item*.** The *item* module allows to describe specific items or products listed on invoices. The term "item" not only refers to a specific traded product, for example a printer, but to anything that can be listed on invoices, for example a working hour. In the purchase-to-pay domain, the term "item" is according to domain experts and EN16931 [10] the preferable one opposed to the term

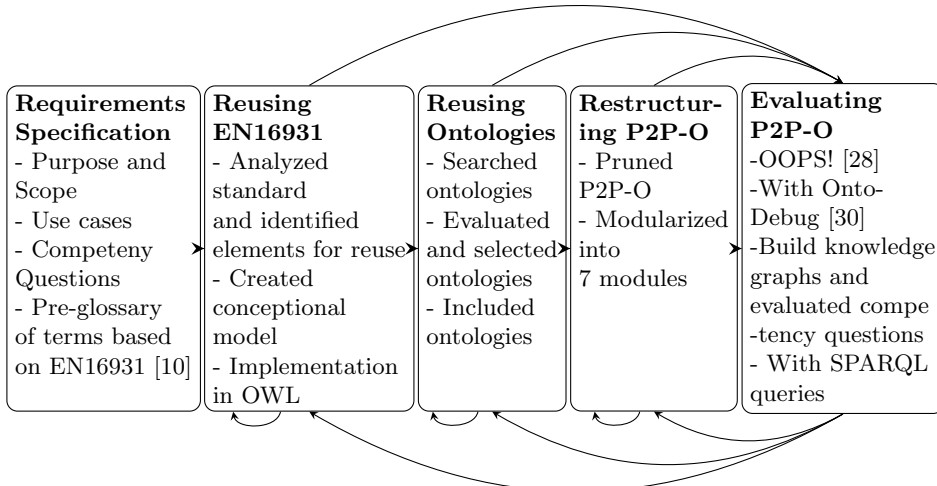

Fig. 2: Methodology for developing P2P-O based on the NeOn Methodology [31]

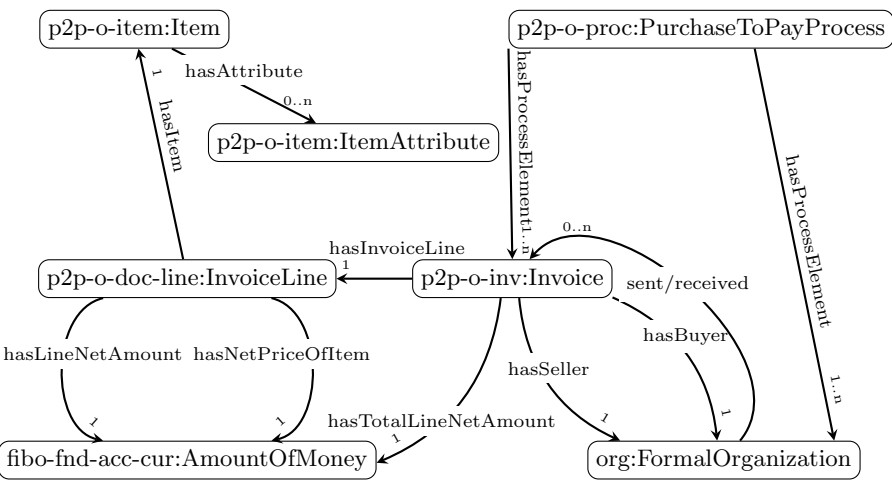

Fig. 3: Excerpt of classes and their relationships in P2P-O. *AmountOfMoney* [3] and *FormalOrganization* [39] are reused.

"product". However, the term "product" is also commonly used interchangeably and therefore included as a synonym for item. According to EN16931 [10], it is only required for an item to have a name, but it can optionally have attributes as well. To easily enable queries such as "retrieve all items on invoices with the colors red and blue" (Tab. 1), an attribute is modeled as a separate class named *ItemAttribute* (Fig. 3). If an item does have such an attribute, this attribute must have one name (e.g. color) and one value (e.g. red) [10].

**Module *price*.** With the *price* module, it is not only possible to describe prices of items but also to make statements about monetary amounts on invoices, such as the total amount with value added tax. Consequently, the class *AmountOfMoney*, which is reused from FIBO [3], is extensively used across P2P-O (Fig. 3). In contrast to EN16931's implications, we have decided not to link the *currency*-property directly to the *Invoice*-class but to the *AmountOfMoney*-class. In our view, this is semantically more appropriate, opens opportunities for reuse and includes the flexibility for stating that the amount of the total value added tax might be in another currency. All other instances of *AmountOfMoney* in the same invoice must still link to the same currency [10].

**Module *documentline*.** The *documentline* module is responsible for lines or positions on a document. As a core function, it enables to make statements about prices and quantities of items listed on documents. Therefore, it imports the modules *item* and *price*. As implied by EN16931 [10], a document line in P2P-O has exactly one item, which is expressed by the *hasItem*-property (Fig. 3). The assured one-to-one relationship between *DocumentLine* and *Item* is essential to relate statements on document line level unambiguously to a corresponding item. Prices about items are made in this module rather than in the *item* module. This allows the expression of more than one price for an item and the traceability of prices to the context of a transaction.

**Module *organization*.** The *organization* module is for describing organizations participating in purchase-to-pay processes. It heavily reuses vocabularies from other ontologies because adequate solutions for our purposes already existed. For instance, this module includes *FormalOrganization* from The Organization Ontology [39] and *Address* from the vCard Ontology [40]. The *BusinessRelationship*-class is intended to describe a dyadic business relationship. Organizations are linked to a business relationship via the *hasCustomer*- and *hasSupplier*-property. *BusinessRelationship* is not implied by EN16931 [10] but introduced in P2P-O. It is useful to make statements about typical characteristics of a business relationship, such as customer or supplier numbers. These are indeed specified by EN16931, but on the document level [10].

**Module *document*.** The *document* module provides classes and properties that are essential for purchase-to-pay documents in general. For more granular

vocabulary regarding invoices, which are special kinds of documents, we created a separate module. A document could have been sent or received by an organization. In P2P-O this is expressed by the properties *sent* and *received*. Because information about organizations is needed, the *organization* module is imported. Cross-references between documents can be expressed by using the *references* object property.

**Module *invoice*.** Because of the focus of P2P-O on invoices, the *invoice* module is the largest sub-module. It imports the modules *document* and *documentline* directly and therefore all other previously introduced modules indirectly as well. It extends the taxonomy of the *document* module and provides more granular vocabulary for kinds of invoices such as *E-FinalInvoice* and *E-PartialInvoice*. To distinguish credit notes from ordinary invoices, *E-CreditNote* and *E-CommercialInvoice* are made disjoint. Therefore, for ordinary invoices, we recommend to use the class *E-CommercialInvoice* instead of the class *E-Invoice*.

Furthermore, this module implements all constraints an invoice must satisfy according to EN16931 [10]. For instance, it must have exactly one seller and one buyer. Particular total amounts are mandatory, like the amounts with and without value added tax. Not mandatory information, such as the allowance amount, are provided as well. An invoice also needs to have at least one instance of the class *InvoiceLine* (Fig. 3), which is modeled as a subclass of *DocumentLine*. In P2P-O this is expressed by the *hasInvoiceLine*-property. Information on invoice line level can be thus related unambiguously to information on invoice level, such as the seller and buyer.

**Module *process*.** According to process requirements in EN16931 [10], the *process* module provides classes for describing specific kinds of purchase-to-pay processes. These are, for instance, a process in which invoiced items are purchased periodically or the payment amount due is paid in advance. These classes are not modeled as disjoint to each other because an instance of a purchase-to-pay process might fit more than those classes [10]. For example, a process may include both, a paying upfront and a corrective invoicing process. Documents and purchase-to-pay processes are linked via the *hasProcessElement*-property and the inverse property *isProcessElementIn* (Fig. 3). With the properties *followsDocument* and *precedesDocument*, it is further possible to express that a document precedes or follows another document in a process.

### 3.4   Availability and Maintenance

P2P-O is available at `https://purl.org/p2p-o` and its accompanying resources at `https://purl.org/p2p-o#res`. For publishing and documenting the ontology, we used WIDOCO [12] to stick to open standards and best practices. Also, resources for building knowledge graphs with P2P-O are made available (e.g. RML mappings) as well as an invoice generator for test and evaluation purposes. Because P2P-O should be easily reusable and adoptable by organizations, the ontology is published under a business friendly permissive license.

Every module of P2P-O has its own version number. When a module becomes backwards incompatible, which we try to avoid, it will be annotated accordingly. The ontology is maintained by a focus group composed of researchers at DFKI[8] and domain experts from the TRAFFIQX[9] network. In monthly meetings, P2P-O related topics, issues and applications are discussed. Because of this research-industry cooperation, P2P-O and its accompanying resources (e.g. RML mappings) aim to contribute to the adoption of semantic web technologies, especially the adoption in SMEs by considering their particular challenges.

### 3.5   Reusability

Because P2P-O is grounded in the core invoice model of EN16931 [10], it is designed to cover the most common purchase-to-pay processes and invoices, and it is also extendable to more specific information needs. This is essential because information needs in the purchase-to-pay domain can vary depending on the concrete application scenario. For instance, in the manufacturing sector, detailed information on invoiced products is valuable, whereas in the service sector, information in supporting documents of an invoice is more important.

Whereas in the presented use case in Section 4.3 only a small subset of purchase-to-pay documents in the TRAFFIQX network have been lifted up to linked data, the transaction volume alone in this network amounts to 40 million per year. Globally, according to Billentis [22], 55 billion e-invoices have been processed in 2019 and the tendency is rising. Noteworthy, P2P-O not only covers e-invoices but also related documents in respective purchase-to-pay processes. For easier reuse, P2P-O is modularized and documented with WIDOCO [12].

## 4   Evaluation and Use Case

### 4.1   Evaluation Based on Ontology Evaluation Tools

To ensure that P2P-O is aligned with current ontology engineering standards, it was iteratively tested against best practices formulated by OOPS! (OntOlogy Pitfall Scanner!) [28]. This tool was in particular valuable for identifying issues concerning modeling inverse relationships and providing a license. To ensure that P2P-O is correct and consistent even when users extend it, OntoDebug [30] was used to annotate test cases and to debug P2P-O.

### 4.2   Evaluation by Constructing and Querying a Purchase-To-Pay Knowledge Graph

On the one hand, we evaluated P2P-O with real-world invoices from the TRAFFIQX network (see Sec. 4.3), which may not be published. On the other hand, to provide a publicly available data set, we evaluated P2P-O with generated

---

[8] `https://comem.ai`
[9] `https://www.traffiqx.net/en/about-us`

test-invoices from our invoice-generator. With this generator it is possible to produce invoices and credit notes in different syntaxes, such as UBL v2.1[10]. These invoices are inspired by the real ones and are in XML.

To construct purchase-to-pay knowledge graphs for evaluating competency questions (Sec. 3.1), we created tailored RML mappings [7] with the use of the CARML extension[11]. It was additionally used because of its ability to deal with namespaces. Thus, we transformed heterogeneous XML-based invoices into semantic invoices. With provided RML mappings, organizations can immediately start experimenting with P2P-O and building purchase-to-pay knowledge graphs. Figure 4 illustrates a part of a constructed knowledge graph in GraphDB[12]. Shown is an excerpt of two invoices (0815-9923-1-a, 08315-93229-1-a) and their relations to each other. Our RML mappings only assert that, for instance, an invoice line has an item; but because of enabled inference, the inverse property *isItemOf* is also available to traverse and query the knowledge graph. To evaluate competency questions (Tab. 1), we executed respective SPARQL queries. Listing 1.1 illustrates the federated query for retrieving items sold in cities with more than 50.000 inhabitants (competency question C15 in Tab. 1).

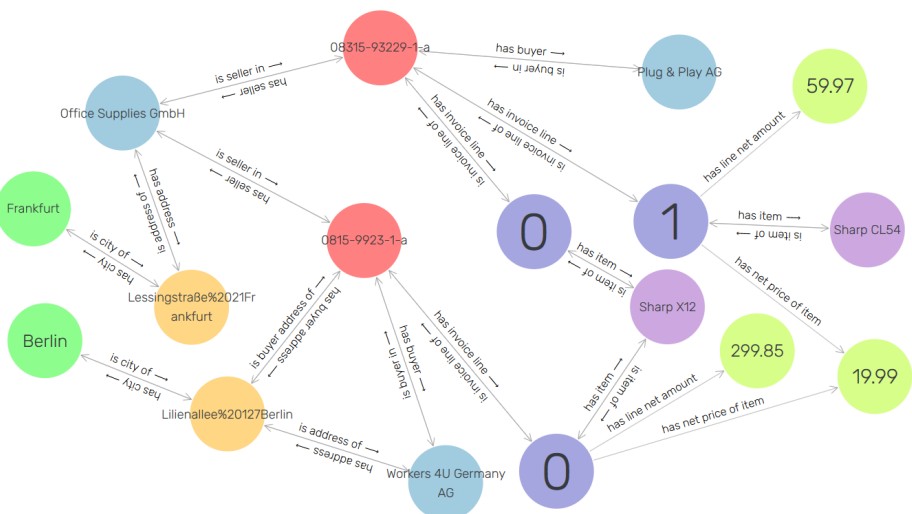

Fig. 4: Screenshot of two synthetic sample invoices in a purchase-to-pay knowledge graph in GraphDB (https://www.ontotext.com/products/graphdb/).

---

[10] https://www.iso.org/standard/66370.html

[11] https://github.com/carml/carml

[12] https://www.ontotext.com/products/graphdb/

Listing 1.1: Federated SPARQL query for items sold in cities with more than 50.000 inhabitants by using P2P-O's vocabulary and DBpedia [2].

```
PREFIX p2po-inv: <https://purl.org/p2p-o/invoice#>
PREFIX p2po-line: <https://purl.org/p2p-o/documentline#>
PREFIX p2po-org: <https://purl.org/p2p-o/organization#>
PREFIX rdf: <http://www.w3.org/1999/02/22-rdf-syntax-ns#>
PREFIX rdfs: <http://www.w3.org/2000/01/rdf-schema#>
PREFIX dbo: <http://dbpedia.org/ontology/>
select * where {
<https://comem.ai/org/Office%20Supplies%20GmbH> p2po-inv:isSellerIn ?inv
    ↪    .
?inv p2po-inv:hasBuyerAddress ?buyerAddress ;
    p2po-inv:hasInvoiceLine ?invLine .
?invLine p2po-line:hasItem ?item ;
        p2po-line:lineInvoicedQuantity ?qty ;
        p2po-line:hasLineNetAmount ?lineNetAmount .
?item rdfs:label ?itemLabel .
?lineNetAmount rdfs:label ?amountLabel .
?buyerAddress p2po-org:hasCity ?city .
?city rdfs:label ?cityLabel .
SERVICE <http://dbpedia.org/sparql> {
?dbCity a ?dbO ;
        rdfs:label ?cityLabel .
?dbCity dbo:populationTotal ?population .
FILTER ((?dbO = dbo:City || ?dbO = dbo:PopulatedPlace ) && ?population >
    ↪    50000) }}
```

### 4.3   Use Case: Applying P2P-O in a Corporate Memory System

After applying artificially generated invoices (Sec. 4.2), we employed real-world invoices from the TRAFFIQX network. This network utilizes an internal intermediate format for their invoices which we exploited to convert 2000 of them into semantic invoices. This enabled the use within a knowledge description layer of a corporate memory such as the DFKI CoMem[13]. CoMem is a corporate memory infrastructure realizing knowledge-based services using enterprise and personal knowledge graphs. With the Semantic Desktop ecosystem, it embeds these services into knowledge workers' working environments [23, 24].

Therefore, P2P-O was published in the CoMem ontology server. A converter was added to the CoMem semantification service which then allowed populating knowledge graphs in CoMem's knowledge base. This enabled the usage of CoMem's existing knowledge services, such as proactive information delivery, semantic search, or ontology-based named entity recognition [20]. Thus, as a use case for a P2P-O-based knowledge service, we implemented a context-specific dashboard within CoMem. With widgets, the user can now grasp context-specific information of an entity, such as items on an invoice (Fig. 5). From those widgets, a user can browse to other resources to see the knowledge space from those resources with other respective context-specific widgets.

---

[13] https://comem.ai

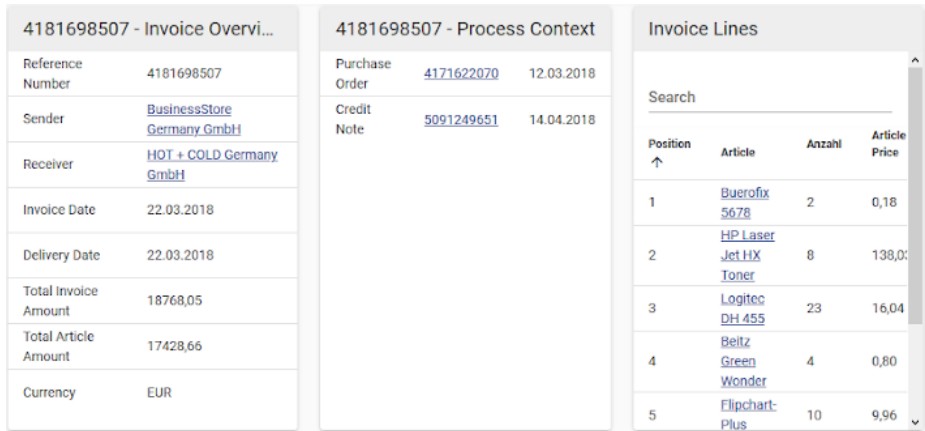

Fig. 5: Excerpt from the context-specific dashboard within CoMem with synthetic sample data.

## 5    Comparison to Related Resources

Semantic technologies in e-procurement, which includes purchase-to-pay processes, are widely recognized for integrating heterogeneous data [29]. In subdomains of e-procurement such as e-awarding [29], ontologies and knowledge services have been successfully applied. For instance, the LOTED2 ontology [8] for tenders. Nečaský et al. [26] present an ontology-based knowledge service for filling out contracts. In purchase-to-pay processes, however, although knowledge work is essential [19], ontology-based knowledge services are largely neglected.

The work of Escobar-Vega et al. [11] addresses semantic invoices in particular. Their approach is to create an ontology out from Mexican electronic invoices with XSL Transformations. Therefore, this approach is rather document-oriented. Resources for creating semantic invoices and the ontology are not publicly available. FIBO [3] is an established ontology in the field of finance with a broad scope. It provides, for instance, vocabulary for loans and investments but not for invoices or purchase-to-pay processes. Likewise, GoodRelations [17] is an established ontology but in the field of e-commerce. It is well-suited to describe offers, products and prices. However, it also does not cover invoice specific vocabulary such as "item" and "invoice line" or a taxonomy for purchase-to-pay documents because this is not in the scope of GoodRelations. The upcoming e-procurement ontology[14] is an endeavor of the European Union towards a uniform vocabulary in the public e-procurement domain. In the current version 2.0.0, the concept of invoice and alike is also missing. Schema.org [16] provides invoice related vocabulary. P2P-O may be able to extend it with more detailed vocabulary necessary for invoice processing as well as with vocabulary that allows to relate invoices to purchase-to-pay processes. Thus, vocabulary is provided that conceives invoices

---

[14] https://github.com/eprocurementontology/eprocurementontology

as part of such processes. The Aggregated Invoice Ontology[15], a dedicated invoice ontology, is not accessible and thus not reusable anymore. However, it was constructed for a pharmaceutical case study and shows the merits and the need of shared invoice vocabulary [13,14].

## 6    Conclusion and Future Work

Owing to the shortcoming of dedicated purchase-to-pay ontologies, this paper presented P2P-O, a modularized ontology which reuses the core invoice model of the European Standard EN16931. To evaluate competency questions, a purchase-to-pay knowledge graph with RML technologies has been constructed and corresponding SPARQL queries executed. In contrast to other contributions, we also provided ready-to-use resources to enable organizations to generate semantic invoices. With a context-specific dashboard, a P2P-O-based knowledge service was illustrated. Its assistance allows an entity-centric view on invoice data by browsing a purchase-to-pay knowledge graph. Future research will focus on enhancing this assistance by applying P2P-O in real-time assistance scenarios and by incorporating other heterogeneous purchase-to-pay data such as business e-mails.

**Acknowledgements** This work was funded by the Investitions- und Strukturbank Rheinland-Pfalz (ISB) (project InnoProm) and the BMBF project SensAI (grantno. 01IW20007).

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
