# OpenReview forum: "P2P-O: A Purchase-To-Pay Ontology for Enabling Semantic Invoices"
_eswc-conferences.org/ESWC/2021/Conference/Resources_Track — ESWC 2021 Resources_

### Official Review · AnonReviewer1 · 2021-01-02
**Good and relevant resource, with some issues to be addressed**

**Rating:** 2
**Confidence:** 4

**Review:**

-- After rebuttal --

I thank the authors for answering some of my comments. Nevertheless, some issues have not been explicitly addressed in their response (ontology versioning and maintenance policy, SPARQL queries, CQ13 and CQ15, lessons learned, links to the "resources for building knowledge graphs with P2P" and the "invoice generator for testa and evaluation purposes"). I expect they will address them in the CR.

As for the domain of the property hasLineNetAmount, I would then suggest to the authors to change Figure 3 accordingly.

My scores remain unchanged.

-- Summary review --
+ The paper is well written and clear
+ The resource is well published and fills a gap in the state of the art
- The ontology has some issues (also wrt consistency)

-- Text --

This paper presents P2P-O, an ontology network for representing electronic invoices and purchase-to-pay processes, based on a European Standard. The resource is well designed, except for some issues, and follows the NeOn methodology.

The paper is well written and clearly explains the state of the art, motivation behind this resource and the resource itself.

- Design

The authors adopt the NeOn methodology (Fig 2). Addressed competency questions, a diagram with core classes and relations and an example of instances modelled according to this ontology are provided.

The resource has been evaluated by using OOPS!, OntoDebug and by verifying competency questions coverage.
Nevertheless, I detected some issues in the ontology.
+ Which is the version of the ontology network? It is not annotated. Also, I would like to know if there is a plan for maintaining the ontologies in terms of their evolution inside the network (e.g. if a change in 1 module affects also another modules, how is it managed?, or can different modules have different version number?)
+ In Section 3.3, the modules of the ontology network are presented. However, at https://purl.org/p2p-o I access an ontology with namespace https://purl.org/p2p-o# that imports (directly or indirectly through the import chain) all other modules. Isn't this an ontology module itself? It seems to me a root module of the network, that returns the whole ontology network. Specific modules have specific URI (e.g. https://purl.org/p2p-o/document#, https://purl.org/p2p-o/documentline#, etc). This should be clarified in the paper. Moreover, I would add a short paragraph explaining the network and the chain of imports between modules.
+ When opening the ontology (from URL https://purl.org/p2p-o) in Protege and running the reasoner (Hermit) I get an inconsistency. Indeed, an axiom (asserted in the root module and in the invoice module) says that an E-invoice (which is a subclass of Document) is related to some string with the property currency. The domain of currency is AmountOfMoney, which is a subclass of QuantityValue. QuantityValue is disjoint with Document. Thus, the inconsistency. The authors should fix this, possibly by removing the axiom, since in the paper they say that an invoice is not directly related to a currency (Section 3.3).
+ Another possible inference is not convincing. The property hasLineNetAmount has DocumentLine as domain. In Figure 3, an Invoice is related to an AmountOfMoney with hasLineNetAmount. In this way, an invoice would be inferred as a DocumentLine too.
+ It is not clear to me which is the semantic difference between hasLineNetAmount and hasNetPriceOfItem.

Both in the figures and in the text it would be useful if you added the module prefix for classes and properties, in order to know which modules (or external ontologies) they belong to.

I am curious about CQ13: what does it mean and how it is addressed in the ontology?
Wrt CQ15, it seems to me way more specific than other CQs, maybe you can generalise e.g. "with more than an amount of inhabitants" instead of "50.000"?

Finally, is there some lessons learned while developing the ontology? If so, it could be briefly included.


- Availability and Reusability

The ontology is published at a persistent URI (purl), with an associated license. It adopts standards such as RDF(S) and OWL. It is well documented by using WIDOCO (in particular, LODE). However, many classes and properties lack rtfs:comment, they could be integrated in order to support understandability and reusability. It directly (and without imports) reuses some state of the art ontologies (as in Table 2), but it is not clear if there are alignments to other external ontologies and/or vocabularies.

Can you make available (if they are not already online) the list of SPARQL queries generated from the CQs, and link it in the paper?

I did not find where the "resources for building knowledge graphs with P2P" and the "invoice generator for testa and evaluation purposes" are made available (as it is said in Section 3.4). They do not seem linked in the paper.


-- Minor comments --

In the whole paper, the opening quotation mark (e.g. information butler at p.2) is not correct. Please, fix it.

P3, 2.1: advices --> advice, or pieces of advice

P4, 2.2: "On the first level (document level), elements are not further divided (e.g. invoice number) --> this is not clear to me

P4, 2.2: put "[10]" before the dot

Table 1. 50.000 --> 50,000

P5, 3.1: "since it provides established guidelines for this exact scenario" --> what do you mean?

P5, footnote 6: I would put here the text of footnote 8

Add footnotes with the URIs of each module in Section 3.3

P7, P8, P9: "respectively" seems used with a wrong meaning: do you mean "related to" or "referring to"?

P7: "hasCurrency-property": I did not find this property, I guess you refer to the datatype property "currency"

P11: "excerpt of two invoices (red nodes)": Refer to the nodes by their names and not their colours

P13: "the concept of an invoice" --> the concept of invoice



**Anonymity:**

No, I would like my review to be deanonymized.

**Strong Points:**

+ The paper is well written and clear
+ The resource is well published and fills a gap in the state of the art

**Subreviewer:**

I submitted this review.

**Weak Points:**

- The ontology has some issues (also wrt consistency)

---

> ### Author Rebuttal · Authors · 2021-01-30
>
> Dear Reviewer,
>
> thank you very much for your very detailed review, your very valuable remarks, and your kind words for example about the relevance. We really appreciate your in-depth analysis of our contribution. It helps a lot to further develop our work and the manuscript. Thank you!
>
> We also ran into the consistency issue in a part of the ontology. We have taken this as an opportunity to improve our ontology publication policies when changes have been made. Thank you for pointing this out. Also thank you for all your other valuable suggestion, we will act upon them (and partially already have).
>
> --
>
> “It is not clear to me which is the semantic difference between hasLineNetAmount and hasNetPriceOfItem.”
>
> hasLineNetAmount refers to the total amount of an invoice line (e.g. 5 packages of Item x which costs y, line net amount = 5y) (without VAT but with allowances/charges on invoice line level)
>
> hasNetPriceOfItem, in this example, would refer to y, so the price of an item (excluding VAT, including discounts)
>
> We have decided to choose DocumentLine as the domain of hasNetPriceOfItem because 1) the price of an item within an invoice line is always the same and 2) in this way it is possible to provide the “invoice context” of a price statement about an item. We see clearly that the annotations with identifiers BT-131 and BT-146 are not sufficient here. As also suggested, we will include rdfs:comments there as well as in other places where they are currently missing.
>
> --
>
> “P5, 3.1: "since it provides established guidelines for this exact scenario" --> what do you mean?”
>
> We mean that our setting (the core invoice model in EN16931 as the main non-ontological resource) fits well with the covered scenarios in NeOn).
>
> --
>
> “Isn't this an ontology module itself? It seems to me a root module of the network, that returns the whole ontology network.”
>
> Yes, you are absolutely right, and internally we refer to this module as “p2p-o-main” (to distinguish it from p2p-o, the ontology). We have discussed this in advance, and at that time we concluded to speak of seven modules in the paper because seven substantial modules with invoice/purchase-to-pay related content are presented (in order to avoid confusion). But maybe it is indeed better to make it more clear in the paper and to mention the root module explicitly. Thank you for pointing this out.
>
> Thank you again for your time and effort spent in preparing this detailed review and for your many other very valuable suggestions and remarks on which we will act upon. Also, thank you for your minor comments with respect to language to improve the manuscript.
>
> Best regards
>
> Michael Schulze (corresponding author)

---

### Official Review · AnonReviewer2 · 2021-01-11
**A resource using the NeOn Methodology to descrive an represent machine-readable electronic invoices**

**Rating:** 1
**Confidence:** 4

**Review:**

Summary
========
This resource paper proposes an ontology following the NeOn methodology to describe and represent electronic invoices in a machine-readable format. The authors use mainly the European Standard EN 16931-1 for their proposal. The ontology is published on a stable PURL with access to the modules of the ontology. However, It was not possible to assess the RML mappings described in the paper, as well as the KG generated and stored in a GraphDB instance. Additionally, the SPARQL queries corresponding to CQs are not available for assessment.
 I understand that this work could help SMEs to start using/generating semantic invoices. However, I miss in the current version of the paper some key differentiator points with the work of [14], even if the authors mentioned that the ontology is not available. I argue that they could have contacted the authors to double-check if their resources (more than 90+ CQs, ontology) could be reused in this paper. The resource is not really a break new ground, but a nice resource in the finance domain in general. However, it does not add much value to existing works but does implement an EU standard lying in some pdf files. One can ask if there are no other standards, for example in the US or other regions that deal with invoice processing.

On the potential impact
====================
The resource does not break new ground. However, the overall goal of increasing the adoption of e-invoices is far interesting. There is a need to have a machine-readable invoice. However, I do not understand why the authors do not build their work on top of existing works, such as the work in [14], which already contains many more CQs. It is clear that the novelty is the modeling using a given standard. However, I do not understand why a previous similar work in [14] is not reused at all, nor mentioned in the modeling process.  As I mentioned previously, the resource is of interest to the financial domain or those applying SemWeb to finance in Europe. However, I’m not sure of an immediate impact of the resource in supporting SemWeb technologies, since this is not clearly identified in the paper.
The resource lacks reusability by people outside of the ones who created the ontology. Nevertheless, it shows an application in the finance domain and purchase.

Reusability
========
There is not a clear evidence of usage by a wider community beyond the resource creators. This is one of my main concerns along with the evaluation process described in the paper. The ontology is not multilingual nor mentions in the future works something along the lines when we know that there are other official languages in Europe. The resource is not general enough but follows a European standard. I wonder if the authors checked if there are other standards ISO, etc in the invoice domain. The resource can be extended according to the authors in Section 3.5. Although the paper stated that “the purchase-to-pay domain can vary depending on the concrete application”, there are no further guidelines for doing such extensions, even for the pharmaceutical scenario cited in the related works. Section 4.3 describes a use case scenario with a converter added in the CoMem ontology server with a context-specific dashboard. Again, it is difficult to assess the use case, and there is no evaluation of the added-value of using semantics for building the dashboard of Figure 5. The authors explain clearly the methodology, the decisions to use a class or a property differently from the standard.

Availability
=========
The resource (at least the ontology) is published at a persistent PURL, even if the final location is a homepage. The resource uses MIT licence, and available online, at least the modules of the ontology. There is a sustainability section to maintain the resource mainly from the experts and the group proposing the ontology. I wonder if there is a Github repository/ mailing-list or any other type of community for any SME to submit issues for updating the resource. The ontology uses OWL and the mappings are generated with RML.

Questions/suggestions
===================
* P. 2: Why is it relevant to know the population of the cities where a company sells the products? Any business reason for SMEs?
* P.5: I do not see the answers to the CQs. Any link to assess them?
* P.5: Please, clarify if there is a hub of e-invoices to be reused by SMEs. How many iterations and cycles were necessary to achieve the development of the ontology? Please, share the lessons learned and/or your experience with the experts.
* P.6: In section 3.3, it could be great to know for stats for each submodule. Why using FIBO for modeling usage note property instead of any other in this list for example https://lov.linkeddata.es/dataset/lov/terms?q=usage+note?
* P.8: I suggest also to show in Figure 3 the modules and their interactions. In which module you call all the other modules? You mention the BusinessRelationship but it is not depicted in the Figure.
* P.9: IMHO, there are too many text descriptions. I miss some listings of the axioms used to model some of the relevant classes. It is not clear in the module process, how you deal with temporality. You do not reuse any time ontology?
* P.10:  How do you use ontodebug for the evaluation?
* P.11: How is UBL related to the ontology described in the paper?  In Figure 4, how is the policy of creating the URIs of the instances? Could you add a link to the original invoices or to the GraphDB instance (if it is public of course).
* P.12: I suggest also have access to the mappings, the SPARQL queries.
* P.13: In [14], there are different of CQs (workflows, business rules, roles, multilingualism, etc) with more than 50 CQs. I still do not understand why that resource is not used for comparison.
* I would suggest the authors to register the resource in a community ontology hub such as LOV or any other suitable for their need.


 After rebuttal
============
Dear authors,

Thanks for these detailed answers to some points I raised during the evaluation of your paper. I agree with you that this work will largely contribute to the adoption of e-invoices by SMEs. Therefore, I adjust my rating according to the convincing answers received. I hope you'll include the suggestions in the camera-ready paper so that the resource will benefit the community.


**Anonymity:**

Yes, I would like my review to remain anonymous.

**Strong Points:**

* The use of a methodology to model, implement and publish the ontology
* The resource is of interest to the financial domain or those applying SemWeb to finance.
* The resource (at least the ontology) is accessible online with a nice webpage based on the WIDOCO




**Subreviewer:**

I submitted this review.

**Weak Points:**

* I miss a clear evidence of usage by a wider community beyond the resource creators or their project.
* It is difficult to assess the use case, the mappings, and the SPARQL queries of the CQs
* There is no evaluation of the added-value of using semantics for building the dashboard of Figure 5.

---

> ### Author Rebuttal · Authors · 2021-01-30
>
> Dear Reviewer,
>
> thank you for your time and effort in reading our contribution and in providing very valuable comments. Thank you also for pointing out strong points wrt to the applied methodology, ontology documentation and the assessment that the ontology is of interest to SemWeb in Finance. Because of the character limit here, we only can address some of your points.
>
> ---
>
> We kindly like to address your main concern wrt usage by a wider community first. Because we would see P2P-O as an emerging ontology (in terms of the ESWC’s CFP), we focused on the alternative way of addressing potential reuse beyond traffiqx that is proposed in the CFP of ESWC. There, it is suggested to address the resource’s potential by quantifying the potential reuse based on other indicators than (re)users. In this regard, in section 3.5, we used as indicators the number of processed e-invoices worldwide (based on Billentis) and numbers within Traffiqx. Also, we addressed/quantified the need for knowledge integration in such processes with numbers from the McKinsey article. We see that we could have made it clearer in the initial submission, and we are eager to improve the clarity in this regard. In this context, we also would like to point out that “end-users” in the Traffiqx network are already very different in terms of industry and organization size. Furthermore, the adoption of e-invoices is, especially for SMEs, an emerging field which begins to gain more and more traction (also due to recent political insistence).
>
> ---
>
> Regarding access to resources:
>
> Resources have been published as static files under the ontology on the main page in the Resource-Section (https://purl.org/p2p-o#res ) (and they haven’t been adjusted since). But we agree with you that these resources were still not easy to see because it was not explicitly linked in the paper (just https://purl.org/p2p-o with the remark that the ontology is published along with these resources).
>
> ---
>
> “The resource is not general enough but follows a European standard. I wonder if the authors checked if there are other standards ISO, etc in the invoice domain.“
>
> In our humble opinion, the main strengths of the Core Invoice Model presented in the reused standard are 1) that experts across industries specified what information on electronic invoices is important for most transactions and which information can be omitted for the sake of such a standard and 2) that the standard also favors a process perspective. We’ve tried to show this in Section 2.2. Therefore, we would argue that such a core invoice model, especially when it is modelled in an ontology, is not restricted to Europe but may be a source for describing invoices worldwide across industries. By being grounded in such a core invoice model, we aimed at being general as much as possible without missing important pieces. Therefore, we would kindly argue that this aspect of generality is a strength of our approach and that this is also one of a key differentiation to the contribution in [14] (which was developed in the context of the pharmaceutical sector). Here we like to emphasis that the eventual configuration of invoices can be depended on the application scenario or industry, which we have tried to show in Section 3.5.
>
> To the best of our knowledge and with consultation with industry experts in the invoicing domain, such a core invoice model in this form is not specified elsewhere. There are indeed national standards, such as SdI (Italy), NAV (Hungary), or FacturaE (Spain) as well as further national standards in south America - which are very advanced - but to the best of our knowledge not in the form of such a unifying core invoice model. A remarkable side remark is, that e.g., Singapore is using this standard (in form of PEPPOL 3.0), but this is actually not our main point we wanted to convey here. We see that in our initial submission the rationale for reusing exactly this standard falls too short, and we are eager to improve it.
>
> ---
>
> P.11: How is UBL related to the ontology described in the paper?
>
> UBL, as well as UN/CEFACT XML Industry Invoice, are valid syntaxes for the core invoice model (kinds of serializations of the invoice model).
>
> ---
>
> “I argue that they could have contacted the authors to double-check if their resources (more than 90+ CQs, ontology) could be reused in this paper.”
>
> Thank you for this great advice. This is a good suggestion we will act upon to systematically check those CQs made for their use case for inclusion (when they fit to the scope and requirements). Also, we are happy to act on the suggestion of contacting the authors to ask 1) whether they have still access to the resources and 2) to make the proposed double check for inclusion (if it is possible to get access to those resources). Thank you for pointing this out.
>
> ---
>
> Thank you again for your very valuable comments and for initiating this scientific discussion.
>
> Best regards, Michael Schulze (corresponding author)

---

> > ### Comment · AnonReviewer2 · 2021-01-30
> > **Convincing answers from the corresponding author**
> >
> > Dear authors,
> >
> > Thanks for these detailed answers to some points I raised during the evaluation of your paper. I agree with you that this work will largely contribute to the adoption of e-invoices by SMEs.
> >
> > ---
> > My point regarding other standards or ISO was to be sure that you were not missing other types of resources in the process of modeling the ontology. If it is clear that the core invoice model could drive any other initiatives, that's actually good news. It means that other initiatives will then reuse this bunch of resources to build any other extensions. Great!
> >
> > ---
> > Regarding the access to resources, please make it clear in the paper their location.
> >
> > Thanks again for your work.

---

### Official Review · AnonReviewer5 · 2021-01-14
**Review of the resource paper: P2P-O: A Purchase-To-Pay Ontology for Enabling Semantic Invoices**

**Rating:** 1
**Confidence:** 4

**Review:**

The authors present their work on developing an ontology for representing entities related to the purchase to pay process.

The paper targets an interesting aspect to improve the extraction and discovery of items and other entities exchanged around invoices.

The authors have well introduced and motivated their work. The design of the ontology has been presented in detail. Evaluation plans have been conducted.

One element that is key for a resource paper and requires attention is to further demonstrate the potential impact of the ontology. The value of the proposed ontology can be better demonstrated for example by showing a potential adoption by users. The dashboard presented was done in a quick way, without highlighting how semantics have played a role in generating this dashboard (i.e., the dashboard looks like simply showing an invoice details). Also the use-case discussed in Section 4.3 was presented with very few details. With respect to the competency question, only one SPARQL query was tested against a question. It is recommended to reduce the number of evaluations (or applications) that were performed from 3 to 1, but present it with enough details to uncover what worked, and what didn't. The way it's currently formulated.

The ontology is made publicly available. It would be valuable to include plans for versioning and maintenance as well.

It would be valuable to make the dataset (or triplestore endpoint) you generated for evaluating the competency questions available to the public so that others in the community can reuse and build on your work.

A suggestion is to aim to align with more online ontology resources (e.g., goodrelations, schema.org).

Minor issues:
- Page 5: C13: "Which items on an invoices have the colors red and blue?" --> Which items on an invoice have the colors red and blue?
- Page 11, Line 16: "competency question" --> competency questions

Good luck with the paper!

**Anonymity:**

Yes, I would like my review to remain anonymous.

**Strong Points:**

Please see main review

**Subreviewer:**

I submitted this review.

**Weak Points:**

Please see main review

---

> ### Author Rebuttal · Authors · 2021-01-30
>
> Dear Reviewer,
>
> thank you very much for your kind words and for your very valuable remarks. Yes, we see that if we would have presented only one evaluation approach, then we could have gone into more detail about this particular one. We also see the merits in doing so, as you pointed out. On the other hand, we would be afraid to miss conveying important evaluation aspects which would be difficult (or not possible) to cover with only one evaluation approach. In this way, we also see merits in providing this threefold approach. But we agree with you on the general line of thought and goal of reducing it (space for further demonstrating potential impact and pointing out “lessons learnt” more detailed). Thank you for pointing this out.
>
> Thank you also for the valuable suggestion to increase the number of SPARQL questions per CQ and the remarks about versioning and maintenance. We will act upon it in a timely manner.
>
> The dataset with example invoices (and more) was published below the ontology (https://purl.org/p2p-o#res , and the data was not adjusted since). It was not easy to see because this section on the page was not explicitly linked back in the initial submission (just https://purl.org/p2p-o with the remark that the data is published along with the ontology). For this, we apologize.
>
> With respect to the online ontologies goodrelations and schema.org, as shortly indicated in the related work section, we plan to align with those ontologies in future versions.
>
> Thank you again for your effort and time spend in reading our contribution and for providing these valuable comments on which we will act upon.
>
> Best regards
>
> Michael Schulze (corresponding author)

---

### Official Review · AnonReviewer3 · 2021-01-15
**well-described ontology with potential impact**

**Rating:** 1
**Confidence:** 4

**Review:**

This is a resource paper that presents an ontology for enabling semantic invoices.

Overall, the paper is well-written, the presentation of the work is very clear and the content of the work seems to be novel.

Considering the review criteria of the Resource track:

**Potential impact**:

All questions related to impact are answered positively. The only one that might not be fully addressed is the following:

- *Has the resource been compared to other existing resources (if any) of similar scope?*

There is a limited comparison in the related work section, but no systematic comparison.

**Reusability:**

All questions related to reusability are answered positively. The only questions that might not have been fully addressed is the following:

-  *Is the resource easy to (re)use? For example, does it have good quality documentation? Are there tutorials availability?*

The vocabulary that is the main contribution is well-documented but the paper refers to a broader set of tools which were used to model and validate the vocabulary, generate the knowledge graph and query it, but the complete solution is not well-document as a reusable workflow.

-  *Is there potential for extensibility to meet future requirements (e.g., upper level ontologies, plugins in protege)?*

The ontology is modular which an explicit clarification regarding extensibility would benefit the paper.

**Design & Technical quality:**

All questions related to design and technical quality are answered positively.

after rebuttal
-------------------
I thank the authors for their effort to answer to the comments. My comment related to the comparison of the work with existing works was not addressed. It was also brought up by AnonReviewer2 along the same lines but it appears that the existing work was not considered nor compared to the work that is proposed in this paper.

**Anonymity:**

Yes, I would like my review to remain anonymous.

**Strong Points:**

- very well-written/documented paper
- novel aspect of an explored domain
- good example of well-thought reuse of vacabularies

**Subreviewer:**

I submitted this review.

**Weak Points:**

a couple of the questions which are defined as criteria for the resource track are not covered, but no major issue.
I think they can be addressed in the camera ready version of the paper (if the paper gets accepted)

---

> ### Author Rebuttal · Authors · 2021-01-30
>
> Dear reviewer,
>
> thank you very much for your thorough review, your kind words and your valuable suggestions. Especially, we appreciate your remark about improving the documentation of the entire reusable workflow, which the ontology is part of. This is a really good idea to focus on. Also, your other suggestions are very valuable so that we will act upon them in a timely manner. Thank you again for taking the time and spending the effort.
>
> Best regards
>
> Michael Schulze (corresponding author)

---

### Official Review · AnonReviewer4 · 2021-01-17
**Semantic invoices ontology. Worth to be presented during the conference.**

**Rating:** 1
**Confidence:** 5

**Review:**


Thank you very much for the response. I've have read it and decided not to change my previous review.

----

The resource presented in this represent a modular ontology for representing invoices semantically. The paper is well written and easy to follow. A brief description of each module is provided and the result has been evaluated in several ways. The methodology and tools used are described.

Main strong points is the fact that the ontology is built in collaboration with a network of stakeholders, in this sense I assume  1) there is already some adoption (though it is not clear whether the data has been provided for the shake of the example or the TRAFFIQX network is really using it within their processes, it would be better to clarify this) and 2) there might be a sustainability plan (again, it is not explicitly mention, please add if so).

The resource fulfill most of the requirements in the resource track: license, permanent URI, documentation, comparison with state of the art, etc.

The only point not mentioned in the paper is whether the ontology is registered in a repository or index. I checked LOV https://lov.linkeddata.es/ (http://www.semantic-web-journal.net/content/linked-open-vocabularies-lov-gateway-reusable-semantic-vocabularies-web-1) and it is not registered. Please submit it.

Another point to improve the reusability of the ontology would be providing a diagram representing a "map" of the modules, that is how they are linked (for example Figure 1 in https://www.w3.org/TR/vocab-ssn/ or http://vicinity.iot.linkeddata.es/vicinity/) and which classes are defined in each module and main relations between them (Figures 3 to 8 in https://www.w3.org/TR/vocab-ssn/). It would be also nice having a dedicated diagram for each module with it detail I see there is a WebVowl visualization provided for https://purl.org/p2p-o but not for each of the modules (for example see http://vicinity.iot.linkeddata.es/vicinity/).

**Anonymity:**

Yes, I would like my review to remain anonymous.

**Strong Points:**

- Potential impact
- In line with ESWC topics
- Design and evaluation

**Subreviewer:**

I submitted this review.

**Weak Points:**

- Registration in ontology indexes or registries
- Documentation for modules

---

> ### Author Rebuttal · Authors · 2021-01-30
>
> Dear Reviewer,
>
> thank you very much for your review and your very valuable comments. Also, thank you for your kind words and for highlighting strong points such as potential impact and design & evaluation. With respect to the collaboration with the network, no, the data is not just only provided for the sake of the example because the setting is as follows that the first author is deeply embedded in the traffiqx network (more precise b4value.net which is the technology provider) and is there treated as a normal (part-time) employer.
>
> We will act on your suggestions in a timely manner (and partially already have). In this regard, to facilitate dissemination and reuse, we agree with you that the ontology should be registered in indexes and registries. According to your suggestions, in the next publishing cycle of the documentation in the next days, we will include WebVowl for the modules to improve their documentation quality. A few days ago (but before the opening of the rebuttal phase), we have already published a basic diagram which depicts the import chain (https://purl.org/p2p-o#imports). Our next steps in this regard are to refine it by also showing the class relations across modules, as you suggested, and by including such diagrams in the sub-modules.
>
> Thank you again for your time and effort spent in providing this valuable feedback and by assessing the resource as worth to be presented during the conference.
>
> Best regards
>
> Michael Schulze (corresponding author)

---

### Decision · Program_Chairs · 2021-02-23

**Decision:**

Accept

**Comment:**

The resource is an ontology network for representing electronic invoices and purchase-to-pay processes, based on a European Standard. The paper is clear and the resource fills a gap in the state of art. Some minor issues have been pointed out in the reviews and have to be considered for the final version.